# Effect of Li^+^ Doping on Photoelectric Properties of Double Perovskite Cs_2_SnI_6_: First Principles Calculation and Experimental Investigation

**DOI:** 10.3390/nano12132279

**Published:** 2022-07-01

**Authors:** Jin Zhang, Chen Yang, Yulong Liao, Shijie Li, Pengfei Yang, Yingxue Xi, Weiguo Liu, Dmitriy A. Golosov, Sergey M. Zavadski, Sergei N. Melnikov

**Affiliations:** 1Shaanxi Province Key Laboratory of Thin Films Technology and Optical Test, School of Optoelectronic Engineering, Xi’an Technological University, Xi’an 710032, China; yangchen931@163.com (C.Y.); lishijie@xatu.edu.cn (S.L.); pfyang@xatu.edu.cn (P.Y.); xiyingxue@163.com (Y.X.); wgliu@163.com (W.L.); 2State Key Laboratory of Electronic Thin Film and Integrated Devices, University of Electronic Science and Technology, Chengdu 610054, China; yulong.liao@uestc.edu.cn; 3Center 9.1 “Electronic Technologies and Engineering Diagnostics of Process Media and Solid State Structures” R&D Department, Belarusian State University of Informatics and Radioelectronics, 220013 Minsk, Belarus; golosov@bsuir.by (D.A.G.); szavad@bsuir.by (S.M.Z.); adminset@bsuir.by (S.N.M.)

**Keywords:** first principles calculation, perovskite, hole transport, doping, ultrasonic spraying

## Abstract

Double perovskite Cs_2_SnI_6_ and its doping products (with SnI_2_, SnF_2_ or organic lithium salts added) have been utilized as p-type hole transport materials for perovskite and dye-sensitized solar cells in many pieces of research, where the mechanism for producing p-type Cs_2_SnI_6_ is rarely reported. In this paper, the mechanism of forming p-type Li^+^ doped Cs_2_SnI_6_ was revealed by first-principles simulation. The simulation results show that Li^+^ entered the Cs_2_SnI_6_ lattice by interstitial doping to form strong interaction between Li^+^ and I^−^, resulting in the splitting of the α spin-orbital of I–p at the top of the valence band, with the intermediate energy levels created and the absorption edge redshifted. The experimental results confirmed that Li^+^ doping neither changed the crystal phase of Cs_2_SnI_6_, nor introduced impurities. The Hall effect test results of Li^+^ doped Cs_2_SnI_6_ thin film samples showed that Li^+^ doping transformed Cs_2_SnI_6_ into a p-type semiconductor, and substantially promoted its carrier mobility (356.6 cm^2^/Vs), making it an ideal hole transport material.

## 1. Introduction

In recent years, organic-inorganic hybrid perovskite solar cells have attracted much attention due to their low preparation cost and high photoelectric conversion efficiency [1,2,3,4], and many methods are still used to improve their photovoltaic performance [5,6,7,8,9]. The maximum conversion efficiency of organic-inorganic hybrid perovskite solar cells is approaching 26% [10]. However, the disadvantages of organic-inorganic perovskite materials like instability in the atmospheric environment and containing the heavy metal Pb, have severely affected the practicability and commercialization of such photovoltaic devices [11,12,13]. Therefore, the research has gradually turned to lead-free inorganic perovskite materials, especially CsSnI_3_ and Cs_2_SnI_6_ [14,15,16,17]. However, these two materials have inherent drawbacks. Our previous research found that CsSnI_3_ was prone to degrade in the atmospheric environment, although it had a suitable band gap and a high optical absorption coefficient [18]. In spite of its relative stability in the atmospheric environment, Cs_2_SnI_6_, which was restricted by the energy level density of the conduction band, had limited optical absorption ability, leading to the low efficiency of photovoltaic devices fabricated based on the Cs_2_SnI_6_ optical absorption layer [19].

The research on Cs_2_SnI_6_ was more focused on using it as a hole transport layer for perovskite and dye-sensitized solar cells. For example, G. Zhang et al. enhanced the hole extraction capability of high–efficiency carbon-based CsPbI_2_Br perovskite solar cells by using Cs_2_SnI_6_ nanocrystals, with the device performance improved from 13.16% to 14.67% [20]. T. T. D. Lien et al. obtained Cs_2_SnI_6_ by natural degradation of CsSnI_3_ in the atmospheric environment and added SnF_2_ in the preparation process to improve its carrier mobility. The mobility of Cs_2_SnI_6_ film treated with 10% SnF_2_ can reach 468.1 cm^2^/Vs [21]. B. Lee et al. added lithium bis (trifluoromethylsulfonyl) imide (Li–TFSI) and 4–tertbutylpyridine (TBP) in undoped n-type Cs_2_SnI_6_, which has been successfully applied as hole transport materials to dye-sensitized solar cells [16].

According to the references and our previous research, intrinsic Cs_2_SnI_6_ is an n-type semiconductor [16,22,23]. If, however, Cs_2_SnI_6_ is obtained by the degradation of CsSnI_3_ or by adding Sn^2+^ ions in the preparation process, it will become a p-type semiconductor [21,24,25,26]. This is because half of the Sn^2+^ in the degradation process will be oxidized to Sn^4+^ and yield isolated [SnI6]^2–^ octahedrons, while the other half will be doped into Cs_2_SnI_6_ [27,28]. However, in the process of forming p-type Cs_2_SnI_6_, Sn^2+^ are easily oxidized by oxygen in the air to generate SnO_2_.

In this paper, Cs_2_SnI_6_ was doped with Li^+^ so that p-type Cs_2_SnI_6_ was directly obtained. Combined with the first principles simulation and specific experiments, the effect of Li^+^ doping on the basic characteristics of Cs_2_SnI_6_, such as energy level structure, electron distribution, optical absorption, and carrier mobility, were systematically studied. The mechanism of Li^+^ doped Cs_2_SnI_6_ forming p-type semiconductor was revealed, and it was confirmed that Li^+^ doped Cs_2_SnI_6_ is an ideal hole transport material.

## 2. Materials and Methods

In this research, the firstprinciples calculation of Li^+^ doped Cs_2_SnI_6_ was executed through the Castep module of Materials Studio software. The simulation was performed using the plane wave ultrasoft pseudopotential method based on density functional theory, and the interaction between electrons and ions was described by ultrasoft pseudopotential. The properties of the object of calculation were calculated based on the optimization of the corresponding model structure (including the relaxation of all atomic positions); the exchange-correlation energy between electrons was described by PBE (Perdew–Burke–Ernzerhof) under the generalized gradient (GGA) combined with LDA+U generalized functions; the band structure, density of states distribution and optical properties were described with HSE06 function; the doping ions concentration was Li^+^:Cs^+^ = 1:4 (supercell 2 × 2 × 2).

The preparation process of Li^+^ doped Cs_2_SnI_6_ thin films by ultrasonic spraying is shown in Figure 1. CsI and LiI in different proportions were dissolved in DMF to form clear mixed solutions, in which the stoichiometric ratios of Li^+^ to Cs^+^ were 1:20, 1:10, 3:20, 1:5, and 1:4, respectively, and the concentration of CsI was 0.2 mol/L. 0.1 mol/L SnI_4_ was added to the mixed solutions and stirred until SnI_4_ was completely dissolved to generate red-brown mixed solutions. Ultrasonic spraying parameters were set as follows: the working distance was 10 cm, the flow rate was 20 μL/s, nozzle moving speed was 5 mm/s, working pressure was 0.2 MPa, and substrate temperature was 130 °C; the prepared mixed solutions were sprayed on the cleaned common glass slides according to the parameters mentioned above 4 times, and then the substrates were baked at 130 °C for 5 minutes after spraying. After that, substrates were immersed in 0.1 g/mL SnI_4_ anhydrous ethanol solution for 1 minute, then rinsed with anhydrous ethanol and dried. Hence the samples were obtained.

Using X-ray diffraction (XRD) (D8 ADVANCE, Bruker AXS, Karlsruhe, German), the phase properties of samples were characterized and analyzed. The field–emission scanning electron microscopy (GeminiSEM 500, Zeiss, Aalen, German) was employed to observe the morphological property of these samples. In addition, Raman testing and analysis was performed under a 532 nm laser for excitation with the laser Raman spectroscope (LabRAM HR Evolution, HORIBA, Fukuoka, Japan). The UV–NIR spectrometer (JASCO V–570 UV/vis/NIR, JASCO, Tokyo, Japan) was used to collect the absorption spectra, and a Hall system (HL 5500, Accent Optical, York, UK) to test the Hall data of these samples.

## 3. Results and Discussion

In this paper, the energy difference of Li^+^ doped Cs_2_SnI_6_ configurations after structural optimization was systematically simulated and calculated for the position of Li^+^ in the Cs_2_SnI_6_ lattice and the stability of Li^+^ doped Cs_2_SnI_6_ lattice. As shown in Figure 2, substitutional doping and interstitial doping might occur in the process of Li^+^ doping Cs_2_SnI_6_, where three different doping models could be constructed, namely Li^+^ replacing Cs^+^ or two types of Li^+^ interstitial doping. According to the three types of doping of Li^+^ in the Cs_2_SnI_6_ lattice, the doping energy difference of each model was calculated in this paper. The calculation formula is as follows:

Li^+^ replacing Cs ^+^:(1)$$\Delta {E=E}_{{\mathrm{Cs}}_{3}{\mathrm{LiSn}}_{2}I_{12}}\;{–\;4E}_{\mathrm{CsI}}\;{–\;2E}_{{\mathrm{SnI}}_{4}}\;{–\;\mu }_{\mathrm{Li}}$$

Li^+^ interstitial doping:(2)$$\Delta {E=2E}_{{\mathrm{Cs}}_{2}{\mathrm{SnI}}_{6}@\mathrm{Li}}\;{–\;4E}_{\mathrm{CsI}}\;{–\;2E}_{{\mathrm{SnI}}_{4}}\;{–\;\mu }_{\mathrm{Li}}$$

The single-point energies of reactants and products are calculated by the first principles, where, $E_{{\mathrm{Cs}}_{3}{\mathrm{LiSn}}_{2}I_{12}}$ and $E_{{\mathrm{Cs}}_{2}{\mathrm{SnI}}_{6}@\mathrm{Li}}$ are the single-point energies of the substitutional doping product Cs_3_LiSn_2_I_12_ and interstitial doping product Cs_2_SnI_6_@Li, respectively. $E_{\mathrm{CsI}}$ and $E_{\mathrm{SnI}4}$ are the single-point energies of reactants CsI and SnI_4_, respectively, and $\mu_{\mathrm{Li}}$ is the chemical potential of Li. The energy difference of each doping type is calculated according to the above formula. The calculation results are shown in Table 1.

According to the above calculation results, it can be seen that the crystal structure is unstable after Li^+^ are doped into Cs_2_SnI_6_ lattice by replacing Cs^+^, hence corresponding compounds fail to be generated in practice. The interstitial doping formed by Li^+^ entering the Cs_2_SnI_6_ lattice has negative energy differences, which indicates that the interstitial doping crystal structures are stable and the target products can be formed in practice. Therefore, Li^+^ can only enter the Cs_2_SnI_6_ lattice in the form of interstitial doping, which is mainly because the radius of the Li^+^ ion is small and it is easy to enter the interstitial voids.

After determining the type of Li^+^ doping, the effect of Li^+^ doping on the electron distribution of Cs_2_SnI_6_ was observed through an electron density difference distribution diagram. As shown in Figure 3a, electron cloud overlap occurs obviously between Sn_4_^+^ and I^−^ in the undoped Cs_2_SnI_6_, with Sn−I bonds forming [SnI_6_]^2–^ octahedrons, among which no electron cloud overlap arises due to the absence of other ions. When a Li^+^ is on interstitial site 1, as shown in Figure 3b, there exists an obvious electron enrichment between the Li^+^ and I^−^, forming a strong interaction. When the Li^+^ is on interstitial site 2, as shown in Figure 3c, there are a few shared electrons between Li^+^ and I^−^, and the interaction is relatively weak. The main reason for that is the distance between Li^+^ and I^−^ on interstitial site 2 are significantly longer than those on interstitial site 1, which weakens the interaction between the Li^+^ and I^−^. This is also the main reason why the energy difference of interstitial site 1 is slightly lower than that of interstitial site 2, and it is easier to yield doping products when the Li^+^ is on interstitial site 1 during the experiment.

In view of this, further analysis was conducted on the energy band and density of states of the configuration of interstitial site 1. As shown in Figure 4a, for the undoped Cs_2_SnI_6_, the band gap is 1.26 eV, and the calculated value is basically consistent with the experimental one. According to the density of states distribution curve, it can be seen that the energy level of the conduction band bottom of Cs_2_SnI_6_ is created by the hybridization of I–p orbital and Sn–s orbital, while that of its valence band top is determined by I–p orbital. In accordance with the band structure diagram of Cs_2_SnI_6_, it can be observed that there is only one energy band at the bottom of its conduction band, and thus the effective density of states is low. According to the Pauli Exclusion Principle, once an electron fills the bottom of the conduction band, it will prevent other electrons from further filling, which is prone to photon absorption saturation. Then, the energy difference between the minimum and sub-minimum conduction band of Cs_2_SnI_6_ is as high as 3.34 eV, making it difficult to effectively absorb photoelectrons. This is also the main reason for the low conversion efficiency of photovoltaic devices based on the Cs_2_SnI_6_ optical absorption layer.

Figure 4b illustrates the band structure and density of states distribution of Li^+^ doped Cs_2_SnI_6_. It is shown that the intermediate energy levels are yielded in the band gap of Li^+^ doped Cs_2_SnI_6_ as a result of the α–spin electrons splitting off from the I–p orbital near the top of the valence band. This is because the introduction of Li^+^ enables a strong correlation between Li^+^ and I^–^, leading to the splitting of the I–p spin-orbital. In addition, the formation of strong interaction between Li^+^ and I^−^ significantly changes the energy band near the bottom of the conduction band, and the introduction of intermediate energy levels effectively improves the electronic density of states of the conduction band, which is conducive to the absorption and transmission of photons, and which can effectively broaden the optical absorption range and enhance the optical absorption ability.

Figure 5 shows the absorption coefficients of undoped Cs_2_SnI_6_ and Li^+^ doped Cs_2_SnI_6_. It can be seen from the illustration that the absorption edges of undoped Cs_2_SnI_6_ and Li^+^ doped Cs_2_SnI_6_ are 1.48 eV and 0.7 eV, respectively, indicating a noticeable red shift. Moreover, the absorption coefficient of Li^+^ doped Cs_2_SnI_6_ considerably increased, which has also proved that the intermediate energy levels improve its optical absorption properties.

As shown in Figure 6, dense thin films with uniform surfaces can be obtained by ultrasonic spraying. According to Figure 6a, cubic phase crystals can be obviously observed on the surface of undoped Cs_2_SnI_6_ thin films, with a grain size of about 500 nm, which is consistent with the theoretical crystalline state of Cs_2_SnI_6_ under normal conditions, namely the face-centered cubic lattice of m 3m point group. When the doping concentration of Li^+^ rise, it can still be clearly observed that the morphology of Cs_2_SnI_6_ remains the cubic phase crystal. However, when the doping concentration reaches more than 20% (Li^+^:Cs^+^ = 1:5), the morphology of the crystals is distorted. Although their cubic structures can still be observed, they are hollow on the inside. XRD test results are required to further determine the specific crystallization.

The XRD spectrums of undoped Cs_2_SnI_6_ and Li^+^ doped Cs_2_SnI_6_ are illustrated in Figure 7. The diffraction peak positions of all samples are identical to the theoretical ones of Cs_2_SnI_6_ standard PDF card # 73–0330 in ICSD (Inorganic Crystal Structure Database). It suggests that Li^+^ doping does not change the crystalline morphology of the face-centered cubic structure of Cs_2_SnI_6_, but there are differences in the relative strength of the diffraction peaks, which may be caused by the morphology change in Figure 6. In addition, there is no impurity diffraction peak but the standard diffraction peaks of Cs_2_SnI_6_ in Figure 7. It indicates that Li^+^ are doped into the lattice of Cs_2_SnI_6_ without changing its inherent crystalline phase, and Li^+^ do not form other lithium compounds. In order to further prove that Li^+^ do not form the corresponding impurities, all samples were tested by Raman spectroscopy.

To further verify that Li^+^ do not form compounds but enter the Cs_2_SnI_6_ lattice, as shown in Figure 8, the Raman spectrums of undoped Cs_2_SnI_6_ and samples of Li^+^ doped Cs_2_SnI_6_ were further studied and analyzed. After spectral fitting, four Raman peaks can be clearly observed. Among them, there are three obvious peaks, located at 76, 90, and 123 cm^−1^, belonging to δ (F_2g_), ν (E_g_), and ν (A_1g_) vibration peaks of Cs_2_SnI_6_, respectively. The tiny peak at 244 cm^−1^ is also the characteristic vibration peak of Cs_2_SnI_6_ as reported in the reference [29]. In addition, there is no other impurity peak, which also confirms the XRD test results that Li^+^ enter the Cs_2_SnI_6_ lattice, and no impurities are produced. Furthermore, as Li^+^ doping concentration increases gradually, the main peak at 123 cm^−1^ shifts, which can be attributed to the change of crystalline morphology brought about by lattice distortion caused by Li^+^ doping.

Figure 9 shows the normalized absorption spectrums of undoped Cs_2_SnI_6_ and Li^+^ doped Cs_2_SnI_6_. As a direct band gap semiconductor, the band gap (*Eg*) of Cs_2_SnI_6_ can be estimated by the classical Tauc relation [21]:*(αhν)^2^ = A (hν − Eg)*(3)
where *hν*, *α* and *A* are photon energy, absorption coefficient, and constant, respectively. The band gap value of the samples can be obtained by extrapolating the straight parts of (*αhν*)^2^ and *hν* curve to the point *α* = 0. According to the inset in Figure 8, it can be directly observed that the band gap value of undoped Cs_2_SnI_6_ is about 1.3 eV, which is almost consistent with that of Cs_2_SnI_6_ reported in the references. It can be found with careful observation that as Li^+^ doping concentration rises, a red shift occurs on the absorption edge and the absorption spectrum range slightly widens, which dovetails with the theoretical calculation results.

In previous research, we found that undoped Cs_2_SnI_6_ featured an n-type semiconductor with an electron mobility of 2.78 cm^2^/Vs [23]. In order to further explore the effect of Li^+^ doping on the electrical properties of Cs_2_SnI_6_, the Hall effect test was conducted on the samples of Li^+^ doped Cs_2_SnI_6_. The test results are shown in Table 2. First, all the Li^+^ doped Cs_2_SnI_6_ samples show p-type semiconductors, and the main reasons are as follows. The simulation results confirmed that Li^+^ enters the Cs_2_SnI_6_ lattice in the form of interstitial doping, which forms a strong correlation between Li^+^ and I^−^, resulting in the electrons as the majority carriers bound by Li^+^. Excessive Li^+^ provides a large number of holes, making Cs_2_SnI_6_ transform from n-type to p-type semiconductors. Second, as the Li^+^ doping concentration is elevated, the Hall mobility of the samples is also substantially enhanced, mainly due to abundant Li^+^, which improves the carrier transport performance of the samples.

## 4. Conclusions

This paper provided a systematic analysis of the effect of Li^+^ doping on the structure, electrical and optical properties of Cs_2_SnI_6_ by firstprinciples calculation, and Cs_2_SnI_6_ samples with different Li^+^ doping concentrations were successfully prepared by ultrasonic spraying. What is more, the simulation results showed that Li^+^ enter the Cs_2_SnI_6_ lattice by interstitial doping, and a strong interaction between Li^+^ and I^−^ was generated accordingly, making the I–p spin-orbital near the top of the valence band split, hence leading to the creation of intermediate energy levels and a red shift on the absorption edge. In addition, the experimental results suggested that doping did not change the crystal phase of the samples, nor produced impurities, and the band gap decreased with the increase of doping concentration, which conformed to the rules of calculation results. The formation of strong interaction between Li^+^ and I^−^ allowed the transformation of Cs_2_SnI_6_ into a p-type semiconductor from an n-type one, and the Hall mobility of the samples was enhanced enormously as Li^+^ doping concentration rose. In summary, this paper uses the simulation calculation and experimental results to analyze the reason why Li^+^ doping changes the conductive type of Cs_2_SnI_6_ and confirms that Li^+^ doped Cs_2_SnI_6_ can be used as an ideal hole transport material in photovoltaic devices.

## Figures and Tables

**Figure 1 nanomaterials-12-02279-f001:**
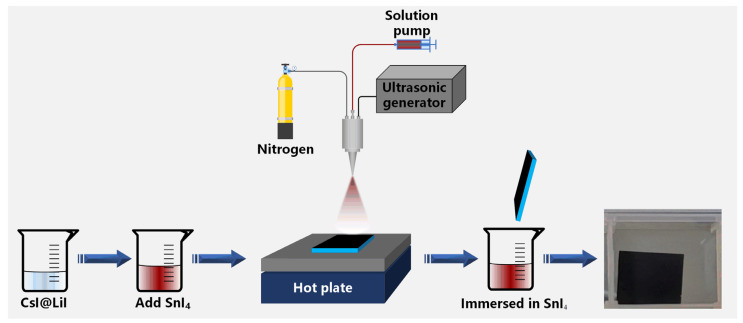
Schematic illustration for the preparation of Li^+^ doped Cs_2_SnI_6_ thin films.

**Figure 2 nanomaterials-12-02279-f002:**
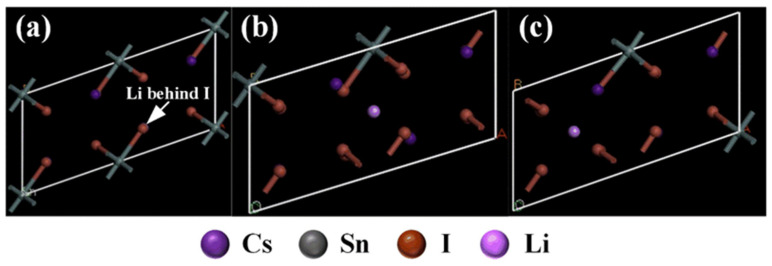
Position of Li^+^ ions in Cs_2_SnI_6_ lattice, (**a**) Li^+^ replaced Cs^+^, (**b**) Li^+^ interstitial site 1 and (**c**) Li^+^ interstitial site 2.

**Figure 3 nanomaterials-12-02279-f003:**
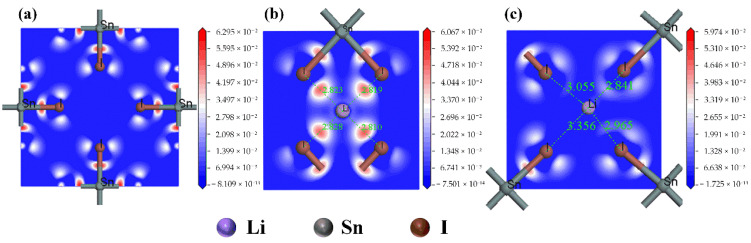
Electronic difference distribution of undoped and Li^+^ interstitial doped Cs_2_SnI_6_, (**a**) undoped Cs_2_SnI_6_, (**b**) Li^+^ interstitial site 1 and (**c**) Li^+^ interstitial site 2.

**Figure 4 nanomaterials-12-02279-f004:**
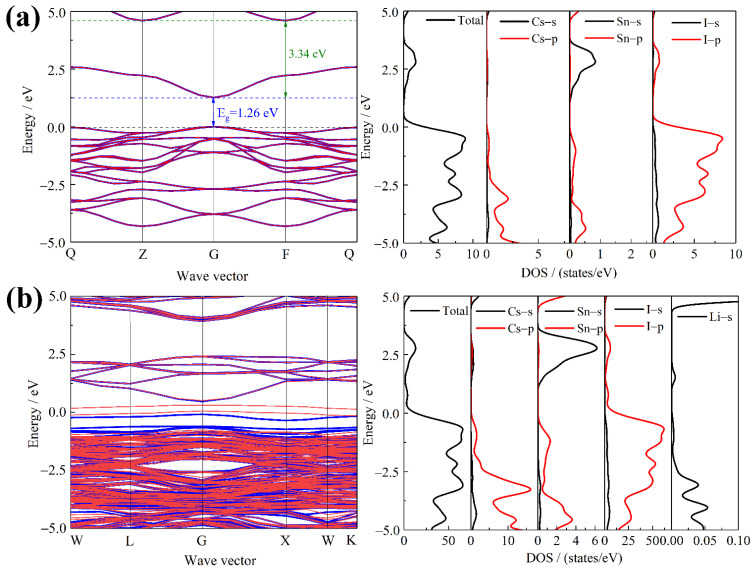
Band structure (red and blue lines are α and β spin polarization, respectively) and state density distribution of undoped and Li^+^ interstitial doped Cs_2_SnI_6_, (**a**) undoped Cs_2_SnI_6,_ and (**b**) Li^+^ interstitial doped Cs_2_SnI_6_.

**Figure 5 nanomaterials-12-02279-f005:**
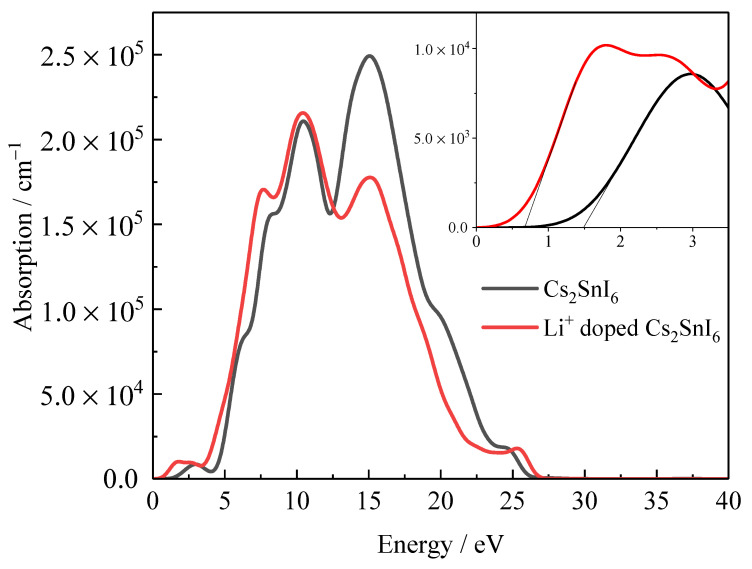
Calculated optical absorption coefficient spectrum undoped and Li^+^ interstitial doped Cs_2_SnI_6_. The inset is the partial enlarged absorption spectra from 0 to 3.5 eV.

**Figure 6 nanomaterials-12-02279-f006:**
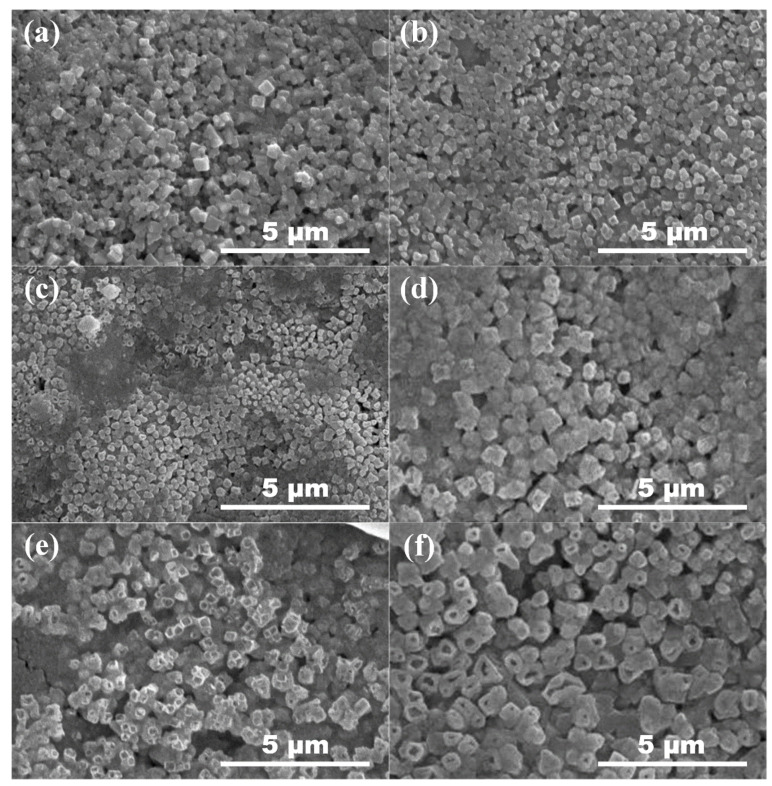
SEM images of undoped Cs_2_SnI_6_ and Li^+^ doped Cs_2_SnI_6_ with different concentrations, (**a**) undoped, (**b**) Li^+^:Cs^+^ = 1:20, (**c**) Li^+^:Cs^+^ = 1:10, (**d**) Li^+^:Cs+ = 3:20, (**e**) Li^+^:Cs^+^ = 1:5 and (**f**) Li^+^:Cs^+^ = 1:4.

**Figure 7 nanomaterials-12-02279-f007:**
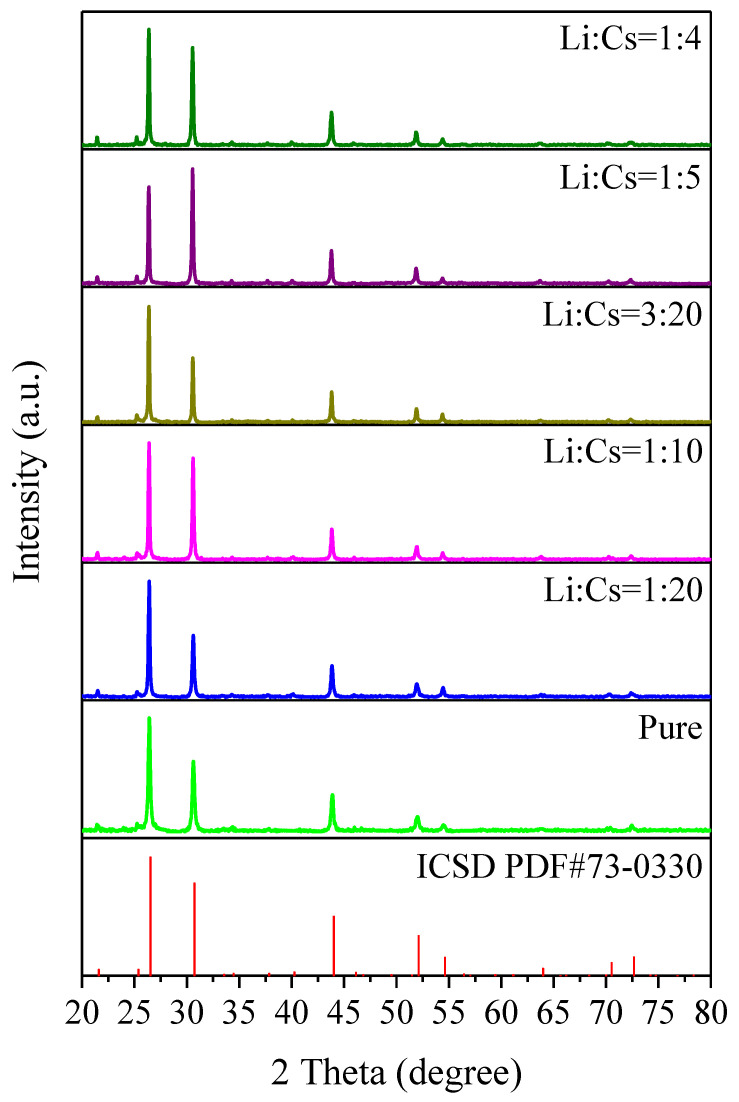
XRD spectrums of undoped Cs_2_SnI_6_ and Li^+^ doped Cs_2_SnI_6_ with different concentrations.

**Figure 8 nanomaterials-12-02279-f008:**
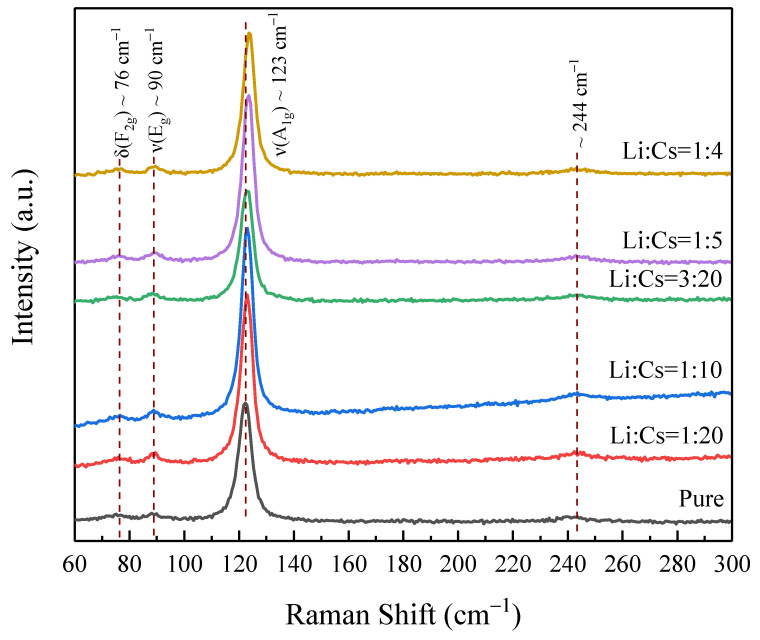
Raman spectrums of undoped Cs_2_SnI_6_ and Li^+^ doped Cs_2_SnI_6_ with different concentrations.

**Figure 9 nanomaterials-12-02279-f009:**
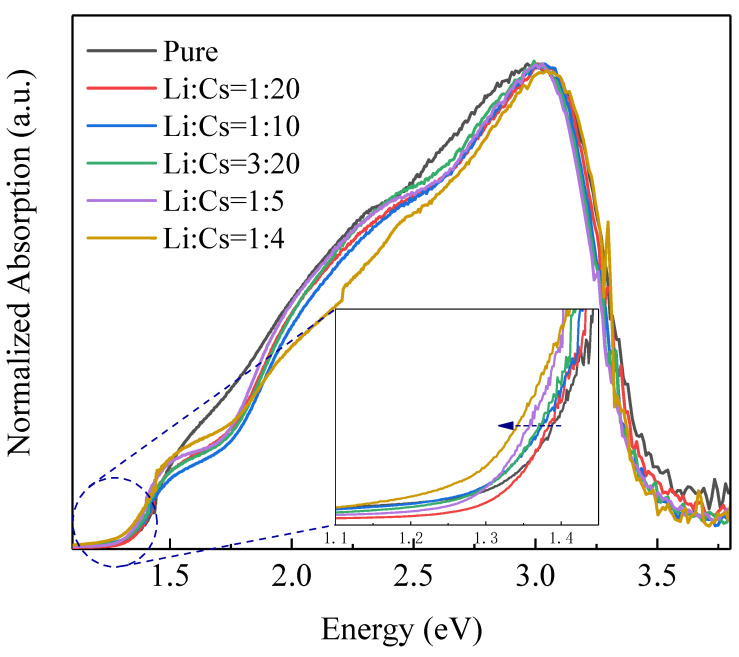
Normalized absorption spectrums of undoped Cs_2_SnI_6_ and Li^+^ doped Cs_2_SnI_6_ with different concentrations. The inset is the partial enlarged normalized absorption spectra from 1.1 to 1.5 eV.

**Table 1 nanomaterials-12-02279-t001:** Energy difference calculation results of Li^+^ ions doped Cs_2_SnI_6_.

Doping Position	Energy Difference
Substitution site	0.0416 eV/atom
Interstitial site 1	−0.01898 eV/atom
Interstitial site 2	−0.01813 eV/atom

**Table 2 nanomaterials-12-02279-t002:** Hall effect data of Li^+^ doped Cs_2_SnI_6_ with different concentrations.

Li^+^:Cs^+^	Conductivity Type	Carrier Concentration (1/cm^3^)	Hall Mobility (cm^2^/Vs)
1:20	p	9.392 × 10^13^	89.51
1:10	p	1.996 × 10^14^	91.89
3:20	p	2.071 × 10^14^	104.5
1:5	p	2.5 × 10^14^	110.45
1:4	p	3.18 × 10^14^	356.6

## Data Availability

Not applicable.

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
