# Peer review of "Effect of Li+ Doping on Photoelectric Properties of Double Perovskite Cs2SnI6: First Principles Calculation and Experimental Investigation"

_nanomaterials, 2022, doi:10.3390/nano12132279_

Round 1

Reviewer 1 Report

The study is devoted to the synthesis and characterization of Li-doped Cs2SnI6 and the calculation of its electron structure upon different level of lithium incorporation.

First of all, the English language must be significantly improved, the reviewer recommends the authors to ask some colleague with sufficient language skills or a native speaker to proofread the manuscript. In present form the document can not be published.

It is shown that the obtained material demonstrates significant level of p-type conductivity, however, not enough data on Hall mobility measurements are provided.

Therefore I can recommend this manuscript to be published in MDPI Nanomaterials only after major revision.

Author Response

The study is devoted to the synthesis and characterization of Li-doped Cs2SnI6 and the calculation of its electron structure upon different level of lithium incorporation.

First of all, the English language must be significantly improved, the reviewer recommends the authors to ask some colleague with sufficient language skills or a native speaker to proofread the manuscript. In present form the document can not be published.

The English languish has now been carefully polished in the revised version as possible as we can. If there is any more suggestion for its betterment further, we will certainly welcome all those for implementation with all appreciation from us.

It is shown that the obtained material demonstrates significant level of p-type conductivity, however, not enough data on Hall mobility measurements are provided.

The main purpose of this paper is not to pursue higher carrier mobility of Li+ doped Cs2SnI6, but to explain the reason why Li+ doped Cs2SnI6 changes its conductive type theoretically, thus confirming that Li+ doped Cs2SnI6 can be used as hole transport materials in photovoltaic devices. In addition, in the modified paper, we add the carrier mobility of pure Cs2SnI6 as we reported before (page 5, line 200). We hope that reviewer can understand our original intention.

In fact, the hall mobility value of p-type Cs2SnI6 reported in this paper is not unprecedented. For example, Thach Thi Dao Lien et al. employed 10% SnF2 to treat Cs2SnI6 thin films and the highest carrier mobility of these thin films is 468.1 cm2/Vs (Tin fluoride assisted growth of air stable transport layer, Mater. Res. Express 2019, 6, 116442). Moreover, Gaurav Kapil et al. also used ultrasonic spraying method to prepare p-type Cs2SnI6 by natural oxidation with SnI2 as raw material in atmospheric environment. Its Hall mobility is as high as 382 cm2/Vs (Investigation of Interfacial charge Transfer in Solution Processed Cs2SnI6 thin films, J. Phys. Chem. C 2017, 121, 13092 – 13100). We will optimize the carrier mobility of Li+ doped Cs2SnI6 in the future work, hoping to further improve.

Therefore I can recommend this manuscript to be published in MDPI Nanomaterials only after major revision.

Reviewer 2 Report

The paper titled Effect of Li+ Doping on Photoelectric Properties of Double Perovskite Cs2SnI6: First-Principles Calculation and Experimental Investigation by Jin Zhang et al requires a deep revision aiming to clearly identify how the author's model reached a so relevant p type mobility, which is hard to be accepted, unless we are dealing with an ideal study case and so, too far from reality. Known data, which are tremendous good can be taken from Jana, S et al in Toward Stable Solution-Processed High-Mobility p-Type Thin-Film Transistors Based on Halide Perovskites, Nov 24 2020 | ACS NANO 14 (11), pp.14790-14797, whose values are more than one decade below the ones now presented. Clarification is required (what is an experiment and what is a simulation) otherwise the paper should not be accepted. Please see the comments below.

Title: It is aligned with the study performed

Abstract: It is misleading once we do not know if the mobility is a theoretical simulated value or if it is an experimental taken value, according to the title. This requires a deep revision and clarification, identifying what is what. As it is, it is misleading.

Introduction: When addressing the present study, it is relevant to see the space charge dynamics that are quite relevant in controlling the transport of carriers generated (see the paper from Shrabani Panigrahi et al in Mapping the space charge carrier dynamics in plasmon-based perovskite solar cells, Sep 14 2019 | JOURNAL OF MATERIALS CHEMISTRY A 7 (34), pp.19811-19819.

Materials and methods: Please identify the conditions of the model used to simulate the results and clarify what is an experiment from it is theoretical data. Concerning sample preparation, how many samples were processed? Can you identify the size and architecture of the structures processed? How reproducible and reliable the process and the structures are? What is the error associated when evaluating structures processed in the same batch but in different spatial locations? What are the errors associated to structure performances from batch to batch? What are the environmental conditions in which the structures were tested?

Did you saw how the Landé g-factors can affect the electron and hole and so the carriers transport anisotropy?

The values depicted in table 2 are just taken under full ideally conditions concerning the of Li in the matrix of the perovskite. Requires deep reflection and clarification on this matter, as already said.

Conclusions: It is missing a summary of the simulation conditions and a better explanation for the enormous p-type mobility values reached. 

References: Need to be updated

Figures: OK

Tables: OK

Author Response

Reviewer 2:

The paper titled Effect of Li+ Doping on Photoelectric Properties of Double Perovskite Cs2SnI6: First-Principles Calculation and Experimental Investigation by Jin Zhang et al requires a deep revision aiming to clearly identify how the author's model reached a so relevant p type mobility, which is hard to be accepted, unless we are dealing with an ideal study case and so, too far from reality. Known data, which are tremendous good can be taken from Jana, S et al in Toward Stable Solution-Processed High-Mobility p-Type Thin-Film Transistors Based on Halide Perovskites, Nov 24 2020 | ACS NANO 14 (11), pp.14790-14797, whose values are more than one decade below the ones now presented. Clarification is required (what is an experiment and what is a simulation) otherwise the paper should not be accepted. Please see the comments below.

Title: It is aligned with the study performed

Abstract: It is misleading once we do not know if the mobility is a theoretical simulated value or if it is an experimental taken value, according to the title. This requires a deep revision and clarification, identifying what is what. As it is, it is misleading.

We are indeed very much grateful to the reviewer for pointing out this issue. First of all, we did not intentionally mislead anyone, the simulation results and test results are clearly stated in the abstract. Second, Hall mobility is the results of testing the experimental samples, which is described in detail in the answers below.

Introduction: When addressing the present study, it is relevant to see the space charge dynamics that are quite relevant in controlling the transport of carriers generated (see the paper from Shrabani Panigrahi et al in Mapping the space charge carrier dynamics in plasmon-based perovskite solar cells, Sep 14 2019 | JOURNAL OF MATERIALS CHEMISTRY A 7 (34), pp.19811-19819.

Based on reviewer’s suggestion, the corresponding content and references have been added to the “Introduction” in the revised version.

Materials and methods: Please identify the conditions of the model used to simulate the results and clarify what is an experiment from it is theoretical data. Concerning sample preparation, how many samples were processed? Can you identify the size and architecture of the structures processed? How reproducible and reliable the process and the structures are? What is the error associated when evaluating structures processed in the same batch but in different spatial locations? What are the errors associated to structure performances from batch to batch? What are the environmental conditions in which the structures were tested?

In the first paragraph of the “Materials and methods”, we have given detailed simulation setting conditions. The purpose of the simulation is to understand how Li+ will be doped into Cs2SnI6 and how Li+ will affect the electron distribution after entering the Cs2SnI6 lattice, which provides a theoretical basis for explaining the transition of the conduction type of Cs2SnI6 after Li+ doping.

According to different doping concentrations, the samples were divided into five groups (Li+ : Cs+ = 1:20, 1:10, 3:20, 1:5 and 1:4). Ten pieces of samples were prepared in each group, among which five pieces were prepared on ordinary sodium-calcium glasses, and the size of each sample was 15 mm Í 15 mm, which was used to test XRD, SEM, Raman and Absorption. The other five pieces were prepared on dedicated Hall effect substrates, as previously reported in J. Mater. Sci. 2018, 53, 378–4386. These too detailed content is not placed in the manuscript, the reason is to avoid too cumbersome description of the manuscript.

The dedicated Hall effect substrates

Ultrasonic spraying is a method for large-scale preparation of thin films, and each group of samples is prepared under the same conditions. Moreover, the test results of XRD, SEM, Raman and absorption are selected under the condition that no special changes are confirmed after multiple tests at different positions. There are five samples for each doping concentration tested by Hall effect. The results in this paper are the average values after five tests, and the error is within 10%.

Cs2SnI6 is relatively stable in the atmospheric environment and has no special requirements for the test environment. In our laboratory, the temperature is maintained at 25±2 oC and the humidity is less than 30%.

Did you saw how the Landé g-factors can affect the electron and hole and so the carriers transport anisotropy?

We have seen relevant literature, but it is helpless that we have not found the corresponding test equipment. We also hope to use the electron spin resonance tester in future research.

The values depicted in table 2 are just taken under full ideally conditions concerning the of Li in the matrix of the perovskite. Requires deep reflection and clarification on this matter, as already said.

We are indeed very much grateful to the reviewer for pointing out this issue. Firstly, the Hall effect test results of Li+ doped Cs2SnI6 were measured by samples, not by simulation, and the carrier mobility cannot be obtained by simulation. The carrier mobility of p-type Cs2SnI6 in this paper (356.6 cm2/Vs) is close to that reported in literatures. Such as Thach Thi Dao Lien et al. employed 10% SnF2 to treat Cs2SnI6 thin films and the highest carrier mobility of these thin films is 468.1 cm2/Vs (Tin fluoride assisted growth of air stable transport layer, Mater. Res. Express 2019, 6, 116442). Moreover, Gaurav Kapil et al. also used ultrasonic spraying method to prepare p-type Cs2SnI6 by natural oxidation with SnI2 as raw material in atmospheric environment. Its Hall mobility is as high as 382 cm2/Vs (Investigation of Interfacial charge Transfer in Solution Processed Cs2SnI6 thin films, J. Phys. Chem. C 2017, 121, 13092 – 13100). It can be seen that the hall mobility value of p-type Cs2SnI6 reported in this paper is not unprecedented, but we just changed a doping material.

Conclusions: It is missing a summary of the simulation conditions and a better explanation for the enormous p-type mobility values reached.

The simulation conditions have been described in detail in the sections of “Materials and Methods”, and will not be repeated in the conclusions. The problem of p-type mobility values has been explained in the reply above.

References: Need to be updated

Necessary references have been added to the revised version.

Figures: OK

Tables: OK

Reviewer 3 Report

The authors reported that the “Effect of Li+ Doping on Photoelectric Properties of Double Perovskite Cs2SnI6: First Principles Calculation and Experimental Investigation” is interesting.

Authors clearly explained how Li+ doping in Cs2SnI6 improving its p-Type character and the absorption properties. The results are well consistent with the characterizations. Moreover, Li+ doping couldn’t change the morphology of pure Cs2SnI6 that is interesting and n-type semiconductor becoming p-Type semiconductor. Such work should be recommended for publication. However, there are certain points which needs to be addressed before proceeding further.

Page 1, line 35: Pd should be Pb I hope this is just a typo mistake.

Page 6, line 181: The sentence should be rephrased as it is bit confusing.

Page 8, Fig. 7: The characteristic peak of Cs2SnI6 at around 13o is missing.

Author Response

Reviewer 3:

The authors reported that the “Effect of Li+ Doping on Photoelectric Properties of Double Perovskite Cs2SnI6: First Principles Calculation and Experimental Investigation” is interesting.

Authors clearly explained how Li+ doping in Cs2SnI6 improving its p-Type character and the absorption properties. The results are well consistent with the characterizations. Moreover, Li+ doping couldn’t change the morphology of pure Cs2SnI6 that is interesting and n-type semiconductor becoming p-Type semiconductor. Such work should be recommended for publication. However, there are certain points which needs to be addressed before proceeding further.

Thanks for the reviewer's positive evaluation. According to the reviewer’s request, the manuscript has been carefully revised.

Page 1, line 35: Pd should be Pb I hope this is just a typo mistake.

We are very sorry for this mistake; the Pd has been modified to Pb in the revised version.

Page 6, line 181: The sentence should be rephrased as it is bit confusing.

According to the reviewer’s request, the sentence has been rephrased in the revised version.

Page 8, Fig. 7: The characteristic peak of Cs2SnI6 at around 13o is missing.

Thanks for the reviewer's suggestion. In fact, when doing XRD tests, we chose the angle from 20 to 80 degrees. The main reason is that we believe that the main diffraction peaks of Cs2SnI6 are concentrated in the range of 20 – 80 degrees, and the diffraction peaks in this range can accurately describe the phase information of Cs2SnI6.

Round 2

Reviewer 2 Report

The paper was properly revised and is now in conditions to be published